# Decentralized Compliance Control for Multi-Axle Heavy Vehicles Equipped with Electro-Hydraulic Actuator Suspension Systems

**DOI:** 10.3390/s25175456

**Published:** 2025-09-03

**Authors:** Mengke Yang, Chunbo Xu, Min Yan

**Affiliations:** 1School of Electrical Engineering, Yanshan University, Qinhuangdao 066004, China; ymk@stumail.ysu.edu.cn (M.Y.); yanmin2021@stumail.ysu.edu.cn (M.Y.); 2School of Information and Control Engineering, Jilin University of Chemical Technology, Jilin 132000, China

**Keywords:** multi-axle heavy vehicles, decentralized compliance control, generalized impedance controller, nonsingular fast integral terminal sliding mode controller

## Abstract

This article introduces a novel decentralized compliance control technique designed to manage the behavior of multi-axle heavy vehicles equipped with electro-hydraulic actuator suspension systems on uneven terrains. To address the challenges of controller design complexity and network communication burden in large-scale active suspension systems for multi-axle heavy vehicles, the decentralized scheme proposed in this paper decomposes the overall vehicle control problem into decentralized compliance control tasks for multiple electro-hydraulic actuator suspension subsystems (MEHASS), each responding to road disturbances. The position-based compliance control strategy consists of an outer-loop generalized impedance controller (GIC) and an inner-loop position controller. The GIC, which offers explicit force-tracking performance, is employed to define the dynamic interaction between each wheel and the uneven road surface, thereby generating the vertical trajectory for the MEHASS. This design effectively reduces vertical vibration transmission to the vehicle chassis, improving ride comfort. To handle external disturbances and enhance control accuracy, the position control employs a nonsingular fast integral terminal sliding mode controller. Furthermore, a three-axle heavy vehicle prototype with electro-hydraulic actuator suspension is developed for on-road driving experiments. The effectiveness of the proposed control method in enhancing ride comfort is demonstrated through comparative experiments.

## 1. Introduction

The interaction between heavy vehicles and road surfaces has garnered significant attention recently. Unlike regular passenger vehicles, heavy vehicles possess high stiffness and large load-bearing capacity. Consequently, the dynamic load fluctuations resulting from their interaction with uneven road surfaces can cause road damage [1,2]. Additionally, the substantial vibration impact significantly compromises the ride comfort of heavy vehicles. Therefore, a primary research focus within the field of heavy vehicle suspension is to optimize the vertical compliance of the suspension system, effectively manage the dynamic interaction between heavy-loaded vehicles and uneven road surfaces, and enhance ride comfort for heavy vehicles [3].

Both leaf spring suspension and hydro-pneumatic suspension (HPS) offer advantages such as high load capacity and a simple structure. Additionally, they can effectively mitigate vehicle body vibrations due to their inherent mechanical damping characteristics. As a result, these suspension systems are widely employed in heavy vehicles [4,5,6]. To comply with road load standards and ensure balanced axle load distribution, heavy vehicles typically increase the number of axles and utilize balanced suspension systems [7]. Currently, leaf spring balanced suspension and interconnected HPS are commonly used in heavy vehicles, with interconnected HPS demonstrating superior vibration reduction capabilities compared to leaf spring balanced suspension [8]. Experimental studies have confirmed that interconnected HPS allows independent control of suspension stiffness and damping characteristics [9,10]. The structure of interconnected HPS includes anti-synchronous interconnections and anti-oppositional interconnections. The judicious application of hydraulically interconnected combinations in heavy-duty vehicles helps alleviate the trade-off between ride comfort and handling stability. However, interconnected HPS faces challenges such as limited suspension dynamic deflection and ride height drift caused by asymmetric stiffness and damping properties [11,12]. To mitigate or eliminate these drawbacks, many researchers have investigated independent active suspension control schemes utilizing electro-hydraulic actuators.

In recent years, significant progress has been made in the field of advanced nonlinear control for electro-hydraulic actuator active suspension systems. Various control approaches, including multi-objective control [13], adaptive control [14], and prescribed performance control [15], have been applied to active suspension, addressing challenges such as parameter uncertainty, nonlinear damping, and nonlinear actuator behavior. Nurkan proposed a backstepping approach for controlling the active suspension of a seven-degrees-of-freedom (DOF) nonlinear full-vehicle model [16]. Na introduced a prescribed performance control method that effectively handles uncertainties and nonlinearities in the active suspension system without relying on function approximators [17]. Sun adopted adaptive robust control for active suspension control equipped with a hydraulic servo actuator, resulting in controllers that are robust against uncertainties in actuator parameters and nonlinearities [18]. Despite the significant advancements in electro-hydraulic active suspension technology, a technical gap remains in developing control schemes for multi-axle heavy vehicles. Multi-axle heavy vehicles equipped with electro-hydraulic actuator active suspension systems exhibit increased controller design complexity and higher CAN bus communication demands compared to two-axle vehicles due to the additional axles. Previous studies have demonstrated that decentralized control architectures can reduce reliance on global communication by enabling localized decision-making [19,20]. Accordingly, this paper aims to develop a decentralized suspension control methodology that decomposes the overall vehicle control problem into individual control tasks for each suspension subsystem. By employing advanced local suspension controllers, the proposed approach reduces controller complexity and alleviates the burden on the CAN bus network. However, existing active suspension control strategies lack precedents for decentralized frameworks.

In the field of robotics control, Hogan introduced impedance control for multi-legged robots [21], which addresses challenges related to hyperstatic structure and compliance interaction. Impedance control is employed to regulate the compliant interaction between the robot and an unknown environment. It has found wide application in manipulator flexible operations and compliance control for multi-legged robots [22,23]. Xukang proposed an adaptive impedance control method to achieve compliant interaction between quadruped wheeled robots and complex terrain environments [24]. Impedance control has also been applied to quadruped robots for optimizing ground reaction force distribution [25]. By enabling independent compliant control of each wheel or leg, these robots achieve compliant interaction with complex terrains, optimizing ground stress distribution across wheels or legs. Yang proposed the concept of mechanical impedance to optimize suspension performance through frequency-dependent equivalent impedance methods [26]; Zhang applied impedance control to full-vehicle active suspension systems, significantly enhancing ride comfort [27]. These approaches provide foundational insights for the decentralized control methodology developed in this work. In light of these considerations, we aim to develop a compliance control scheme for multi-axle heavy-duty vehicles. This scheme actively balances wheel load distribution and provides vertical compliance for the vehicle wheels.

In electro-hydraulic servo active suspension systems, the servo control of electro-hydraulic actuators plays a critical role in determining overall suspension performance. To achieve high-precision control, various strategies, such as adaptive backstepping control [28], sliding mode control [29], and predefined performance control [30], have been employed. Among these, sliding mode control is widely favored for its robustness and reduced reliance on precise system parameters. Furthermore, fast terminal sliding mode control has been developed to mitigate chattering and ensure finite-time convergence [31]. To address the challenges posed by system perturbations and nonlinearities, and to enhance the accuracy of electro-hydraulic servo control for improved suspension performance, this paper proposes a non-singular fast integral terminal sliding mode control (NFITSMC) method.

In this paper, we present a novel decentralized compliance control (DCC) approach for a three-axle heavy vehicle equipped with active electro-hydraulic actuator suspension systems. The objective is to address the overall suspension control problem by dividing it into individual compliance control sub-problems for each electro-hydraulic actuator suspension subsystem (EHASS) in response to road disturbances. The dynamic interaction between each wheel and the rough road surface is managed by a local compliance controller for each EHASS. The compliance control strategy consists of a generalized impedance rule and a position controller. Specifically, the generalized impedance rule, which features explicit force-tracking performance, is used to determine the vertical trajectory for each EHASS. This approach minimizes the force transmitted from the road surface to the chassis, thereby reducing the vibration transferred to the vehicle body. The NFITSMC method is employed to track the predetermined trajectories of the EHASS. Simulations and experiments are conducted under various road inputs to validate the effectiveness of the proposed method.

## 2. Three-Axle Heavy Vehicle Dynamic Model

In this section, we consider a dynamic model of a three-axle heavy vehicle, where the suspension units exclusively consist of electro-hydraulic actuators. To facilitate the controller design process, we divide the 9-DOF vehicle model into two components: the vehicle dynamic model and the multiple electro-hydraulic actuators suspension subsystems (MEHASS) models.

### 2.1. Vehicle Dynamics Model

The 9-DOF model of the three-axle heavy vehicle with nonlinear electro-hydraulic actuators is shown in Figure 1. *Z*, θ, and φ stand for the heave, pitch, and roll motions of the vehicle body, respectively. *M* is the sprung mass, $m_{si}$ and $m_{ui}$, i=1,2,3,4,5,6, are the sprung masses and the unsprung masses of left front, right front, left center, right center, left rear, and right rear, respectively. $z_{ui}$ is the unsprung mass displacement and $q_{i}$ is the road input to the related wheel. $k_{ti}$ and $b_{ti}$ represent the stiffness and damping coefficients of each tire, respectively. $F_{ui}$ represents the output force of the related electro-hydraulic actuators. $F_{fi}$ represents the unmodeled friction force of the related electro-hydraulic actuators. $I_{x}$ and $I_{y}$ denote the mass moment of inertia for the roll and pitch motions, respectively. $l_{a}$, $l_{b}$, $l_{c}$, and $\frac{1}{2}l_{d}$ are the distances of the suspension to the center of the vehicle body mass. *v* is the velocity of the vehicle in the x-direction. The dynamic equations of the three-axle heavy vehicle model with electro-hydraulic actuators are established based on the assumption that the pitch and roll angles are small, as follows:(1)$$\left\{\begin{array}{c} M\ddot{Z}=\sum_{i=1}^{6}\left(F_{ui}-F_{fi}\right)-Mg\\ I_{y}\ddot{\theta }=-l_{a}\sum_{i=1}^{2}(F_{ui}-m_{si}g-F_{fi})+l_{b}\sum_{i=3}^{4}(F_{ui}-m_{si}g-F_{fi})\\ +\left(l_{b}+l_{c}\right)\sum_{i=5}^{6}(F_{ui}-m_{si}g-F_{fi})\\ I_{x}\ddot{\varphi }=\frac{1}{2}l_{d}\sum_{i=1,3,5}\left(F_{ui}-m_{si}g_{i}-F_{fi}\right)\\ -\frac{1}{2}l_{d}\sum_{i=2,4,6}\left(F_{ui}-m_{si}g-F_{fi}\right)\\ m_{ui}{\ddot{z}}_{ui}=k_{ti}\left(q_{i}-z_{ui}\right)+b_{ti}\left({\dot{z}}_{qi}-{\dot{z}}_{ui}\right)-\left(F_{ui}-m_{si}g-F_{fi}\right) \end{array}\right.$$

### 2.2. MEHASS Model

The suspension system of a three-axle heavy-duty vehicle is composed of six independent sets of EHASS. These MEHASS rigidly connected the wheels to the vehicle body. A schematic diagram of the EHASS for an individual wheel, as depicted in Figure 2, consists of an asymmetric hydraulic cylinder and its corresponding electro-hydraulic servo valve.

Neglecting the external leakage, the pressure dynamic equation and flow equation of the *i*th EHASS can be written as follows:(2)$$\left\{\begin{array}{c} {\dot{P}}_{i1}=\frac{\beta_{e}}{V_{01}+A_{1}\Delta z_{i}}\left[Q_{i1}-A_{1}\Delta {\dot{z}}_{i}-C_{t}\left(P_{i1}-P_{i2}\right)\right]\\ {\dot{P}}_{i2}=\frac{\beta_{e}}{V_{02}-A_{2}\Delta z_{i}}\left[-Q_{i2}+A_{2}\Delta {\dot{z}}_{i}+C_{t}\left(P_{i1}-P_{i2}\right)\right]\\ Q_{i1}=k_{q}x_{iv}\left[s\left(x_{iv}\right)\sqrt{P_{s}-P_{i1}}+s\left(-x_{iv}\right)\sqrt{P_{i1}-P_{r}}\right]\\ Q_{i2}=k_{q}x_{iv}\left[s\left(x_{iv}\right)\sqrt{P_{i2}-P_{r}}+s\left(-x_{iv}\right)\sqrt{P_{s}-P_{i2}}\right]\\ s\left(x_{v}\right)=\left\{\begin{array}{c} 1,\;x_{v}\geq 0\\ 0,\;x_{v}<0 \end{array}\right. \end{array}\right.$$
where $\Delta z_{i}$ denotes the *i*th EHASS displacement, $P_{i1}$ and $P_{i2}$ denote the pressures in the non-rod chamber and rod chamber of the *i*th EHASS, and $A_{1}$ and $A_{2}$ are the ram areas of the non-rod chamber and rod chamber, respectively. The forces generated by actuators can be described as $F_{ui}=P_{i1}A_{1}-P_{i2}A_{2}$.

The supply pressure and return pressure of the whole vehicle hydraulic system are denoted by $P_{s}$ and $P_{r}$. $Q_{i1}$ and $Q_{i2}$ are the supplied flow rate to the non-rod chamber and the return flow rate of the rod chamber, respectively. $x_{iv}$ is the servo valve spool displacement of the ith EHASS. An approximated relation between $x_{iv}$ and control input current $u_{i}$ is described as $x_{iv}=k_{v}u_{i}$. Assuming that every EHASS has the same parameters and features, $V_{01}$ and $V_{02}$ denote the initial control volumes of the two actuator chambers. $\beta_{e}$ is the effective oil bulk modulus, and $C_{t}$ is the internal leakage coefficient of the actuator chambers. $k_{q}$ is the flow gain coefficient of the servo valve.

### 2.3. Problem Description

The dynamic model of the multi-axle heavy-duty vehicle and the electro-hydraulic actuators model are presented above. To achieve active suspension control and enhance ride comfort, most existing studies such as those cited in [17,18] employ centralized control schemes. A representative centralized vehicle control architecture is illustrated in Figure 3, which typically includes a centralized attitude controller for the entire vehicle, along with force or position controllers for the actuators within each suspension subsystem. However, as the number of axles increases, the control architecture becomes increasingly complex, and the volume of data transmitted over the CAN bus grows substantially. To address these challenges, this paper proposes a decentralized suspension control approach. The proposed method replaces the complex centralized control structure with a superior local decision-making strategy for each suspension subsystem, thereby reducing the communication load on the CAN bus while maintaining high vehicle ride quality.

## 3. The DCC Method Design

The real-time interaction between a three-axle heavy vehicle and a rough road surface constitutes a highly nonlinear system with a hyperstatic structure. The force applied to the tire from the road has a significant impact on the vehicle body through the suspension system, potentially resulting in road damage and reduced ride comfort in the absence of a compliant suspension system. To address the challenges posed by this large-scale nonlinear system and simplify the controller design, we propose a DCC based on impedance control in this paper. Our method aims to establish compliant interaction behavior for each EHASS by leveraging impedance control in accordance with the tire force.

The DCC method is composed of six individual cascade compliance controllers, as shown in Figure 4. Each EHASS is controlled by its own cascade compliance controller, utilizing local information on tire force to effectively manage the dynamic interaction between each wheel and the rough road surface. The cascade controller consists of an outer-loop GIC and an inner-loop position controller. The outer loop generates the vertical compliance trajectory for the EHASS, while the inner loop achieves tracking control through an NFITSMC.

### 3.1. Generalized Impedance Rule

Impedance control aims to establish a desired dynamic relationship between the end-effector position of a robot and the contact force. In this paper, we employ a position-based impedance control approach to establish the dynamic interaction behavior between tire forces and the vertical trajectory of the MEHASS. The approach for a single wheel is as follows:(3)$$B_{d}\left({\dot{x}}_{d}-{\dot{x}}_{r}\right)+K_{d}\left(x_{d}-x_{r}\right)=F_{d}-F_{ze}$$
where $B_{d}$ and $K_{d}$ represent the desired damping and stiffness parameters, respectively. $F_{ze}$ and $F_{d}$ represent the force applied on the tire from the road and the desired force, and $x_{r}$ and $x_{d}$ represent the reference equilibrium trajectory and the desired compliance trajectory of the EHASS.

Note the following:1.$F_{ze}$ refers specifically to the vertical forces applied on the tire from the road. When a three-axle heavy vehicle is stationary on a level road surface, the vertical force applied to each wheel is equal to the static load force. However, when the vehicle encounters bumps or potholes, the vertical force dynamically increases or decreases.2.Considering the rigid connection between the wheel and the vehicle body through the EHASS, this article simplifies the analysis by disregarding tire compression and directly linking the tire force error to the vertical displacement of the EHASS for impedance control.

Ensuring real-time stable contact between the wheels and uneven road surfaces is crucial for handling stability and ride comfort of a vehicle during driving. Thus, when controlling the vehicle’s suspension, it becomes necessary to satisfy two requirements regarding the dynamic tire force between the wheels and the road surface. Firstly, considering the vehicle’s load capacity, it is essential to ensure the static load capacity of each wheel and maintain a balanced load distribution among all wheels. Secondly, the goal is to minimize dynamic loads to achieve optimal vibration reduction effects. To meet these requirements, we propose the use of generalized impedance control based on position control. The standard static load of each wheel is set as the desired force, while the dynamic tire force serves as the external force to design a vertical compliant trajectory for the wheel.

The DCC method for a three-axle heavy vehicle, based on the generalized impedance formula that relates position error to force tracking error, is formulated as follows:(4)$$B_{di}\left({\dot{x}}_{di}-{\dot{x}}_{ri}\right)+K_{di}\left(x_{di}-x_{ri}\right)=-K_{fi}e_{fi}-K_{Ii}\int_{0}^{t}e_{fi}d_{t}$$
where $e_{fi}=F_{zei}-F_{di}$. $K_{fi}$ and $K_{Ii}$ indicate the force tracking parameters. Generalized impedance control overcomes the limitation of traditional impedance control methods that fail to explicitly track contact forces, thereby improving the force tracking performance of impedance control. The stiffness coefficient $K_{di}$ determines the softness and hardness of the suspension system and can be adjusted to enhance vibration reduction capacity. The damping coefficient $B_{di}$ determines the dynamic performance of the suspension system and can be adjusted to reduce actuator overshoot suppress oscillation. $K_{fi}$ and $K_{Ii}$ determine the tracking performance of the suspension system for static load-bearing forces. When selecting parameters, a reasonable compromise should be made between impedance parameters and force tracking parameters to ensure the maximum vibration isolation capacity of each suspension subsystem and the rapid expansion and rebound stroke of the suspension actuator.

By comprehensively adjusting the stiffness parameters, damping parameters, and force tracking parameters in Formula (Equation 4), the vertical compliant trajectories of the MEHASS with force tracking performance can be obtained. These trajectories ensure stable and balanced load-bearing forces for each wheel, isolate road surface vibrations, and improve ride comfort. The trajectory tracking controller for the MEHASS is presented in the following subsection.

### 3.2. Nonsingular Fast Integral Terminal Sliding Mode Position Controller

The accuracy of the inner-loop position control is pivotal in determining the overall performance of the DCC method. Therefore, this section presents the proposed design for the position control of the MEHASS. The MEHASS in the three-axis heavy vehicle are characterized by strong nonlinearity, model uncertainty, and uncertain external disturbances. To achieve exceptional tracking performance and high-performance electro-hydraulic servo position control, this paper employs an NFITSMC based on an extended state observer. This control approach is utilized to accomplish precise position tracking control of the MEHASS.

The position control model of the MEHASS is as follows:(5)$$m_{si}\Delta {\ddot{z}}_{i}=F_{ui}-m_{si}g-f_{i}\left(\Delta z_{i},\Delta {\dot{z}}_{i}\right)+d_{i}$$
where $F_{ui}=P_{i1}A_{i1}-P_{i2}A_{i2}$, and $f_{i}\left(\Delta z_{i},\Delta {\dot{z}}_{i}\right)$ and $d_{i}$ represent the unmodeled friction forces and modeling uncertainties.

In order to facilitate controller design, a single EHASS is used for the position controller design. The position control model of the EHASS is reduced based on the methodology described in reference [32]. By defining the state vector as $x=\left[x_{1},x_{2}\right]=\left[\Delta z,\Delta \dot{z}\right]$, the second-order state equation of the EHASS can be obtained as(6)$$\left\{\begin{array}{c} {\dot{x}}_{1}=x_{2}\\ {\dot{x}}_{2}=f+b_{0}u\\ y=x \end{array}\right.$$
where $f=\frac{-m_{s}g-f\left(x_{1},x_{2}\right)+d}{m_{s}}+\left(b-b_{0}\right)u$, $bu=\frac{P_{1}A_{1}-P_{2}A_{2}}{m_{s}}$, in which $b_{0}$ is linear estimation of virtual control gain *b* for electro-hydraulic servo systems. *f* represents integration terms for disturbances, including unmodeled friction forces and modeling uncertainties.

The lemmas for designing the position controller are as follows:

**Lemma** **1.**

(7)
$$\begin{array}{c} {\dot{\varepsilon }}_{0}=-\lambda_{0}L^{\frac{n}{n+1}}\left|\varepsilon_{0}\right|\mathrm{sign}\left(\varepsilon_{0}\right)+\varepsilon_{1}\\ {\dot{\varepsilon }}_{1}=-\lambda_{1}L^{\frac{n-1}{n}}\left|\varepsilon_{1}-{\dot{\varepsilon }}_{0}\right|\mathrm{sign}\left(\varepsilon_{1}-{\dot{\varepsilon }}_{0}\right)+\varepsilon_{2}\\ \vdots \\ {\dot{\varepsilon }}_{n-1}=-\lambda_{n-1}L^{\frac{1}{2}}\left|\varepsilon_{n-1}-{\dot{\varepsilon }}_{n-2}\right|\mathrm{sign}\left(\varepsilon_{n-1}-{\dot{\varepsilon }}_{n-2}\right)+\varepsilon_{n}\\ {\dot{\varepsilon }}_{n}=-\lambda_{n}L\mathrm{sign}\left(\varepsilon_{n}-{\dot{\varepsilon }}_{n-1}\right)-\rho \left(t\right) \end{array}\;$$

*is a finite-time stable system, where $\varepsilon_{i}$ is the system state variable, $L>0,\;\lambda_{i}>0$, and $\left|\rho (t)\right|\leq L$ [33].*


**Lemma** **2.**
*Considering the system $\dot{x}=f\left(x\right)$, $x\left(0\right)=x_{0}$, where $x\in R^{n}R$, $f\left(x\right):R^{n}\to R$ is continuous on $R^{n}$ and f(0)=0. Supposing there exists a continuous positive definite Lyapunov function V(x) that satisfies the following: $\dot{V}\leq -\alpha V^{p_{1}}-\beta V$, where $\alpha ,\;\beta >0,\;p_{1}\in \left(0,1\right)$. Then, the system can realize finite-time convergence; the settling time is bounded by $T\leq \frac{1}{\beta (1-p_{1})}\mathrm{In}\left(\frac{\alpha +\beta V_{0}^{1-p_{1}}}{\alpha }\right)$ [31].*


**Lemma** **3.**

(8)
$$\begin{array}{c} {\dot{\eta }}_{1}=\eta_{2}\\ {\dot{\eta }}_{2}=\eta_{3}\\ \ldots \\ {\dot{\eta }}_{n-1}=\eta_{n}\\ {\dot{\eta }}_{n}=c_{n}\mathrm{sgn}\left(\eta_{n}\right){\left|\eta_{n}\right|}^{\alpha_{n}}-\ldots -c_{1}\mathrm{sgn}\left(\eta_{1}\right){\left|\eta_{1}\right|}^{\alpha_{1}} \end{array}\;$$

*is a finite-time stable system, where $c_{i}$ and $\alpha_{i}$ are positive constants. $c_{i}$ is determined to guarantee that the polynomial $p^{n-1}+c_{n-1}p^{n-2}+\ldots c_{2}p+c_{1}$ is Hurwitz [34].*


The following finite-time disturbance observer (FDO) is designed to observe and compensate for disturbance integration terms of the EHASS:(9)$$\begin{array}{c} {\dot{z}}_{0}=b_{0}u+v_{0}\\ v_{0}=-\lambda_{0}L^{\frac{1}{3}}{\left|z_{0}-x_{2}\right|}^{\frac{2}{3}}\mathrm{sign}\left(z_{0}-x_{2}\right)+z_{1}\\ {\dot{z}}_{1}=v_{1}\\ v_{1}=-\lambda_{1}L^{\frac{1}{2}}{\left|z_{1}-v_{0}\right|}^{\frac{1}{2}}\mathrm{sign}\left(z_{1}-v_{0}\right)+z_{2}\\ {\dot{z}}_{2}=v_{2}\\ v_{2}=-\lambda_{2}L\mathrm{sign}\left(z_{2}-v_{1}\right) \end{array}$$
where $z_{0}$, $z_{1}$, and $z_{2}$ are the estimates of $x_{2}$, *f*, and $\dot{f}$, and the parameter *L* satisfies L>0 and $\left|\ddot{f}\right|\leq L$.

Denoting ${\tilde{z}}_{0}=z_{0}-x_{2}$, ${\tilde{z}}_{1}=z_{1}-f$, ${\tilde{z}}_{2}=z_{2}-\dot{f}$ are the observation. By taking the derivative of observation errors, it can be concluded that(10)$$\begin{array}{c} {\dot{\tilde{z}}}_{0}=v_{0}-f\\ =-\lambda_{0}L^{\frac{1}{3}}{\left|z_{0}-x_{2}\right|}^{\frac{2}{3}}\mathrm{sign}\left(z_{0}-x_{2}\right)+z_{1}-f\\ =-\lambda_{0}L^{\frac{1}{3}}{\left|{\tilde{z}}_{0}\right|}^{\frac{2}{3}}\mathrm{sign}\left({\tilde{z}}_{0}\right)+{\tilde{z}}_{1} \end{array}$$
and(11)$$\begin{array}{c} {\dot{\tilde{z}}}_{1}={\dot{z}}_{1}-\dot{f}\\ =-\lambda_{1}L^{\frac{1}{2}}{\left|z_{1}-v_{0}\right|}^{\frac{1}{2}}\mathrm{sign}\left(z_{1}-v_{0}\right)+z_{2}-\dot{f}\\ =-\lambda_{1}L^{\frac{1}{2}}{\left|z_{1}-v_{0}\right|}^{\frac{1}{2}}\mathrm{sign}\left(z_{1}-v_{0}\right)+{\tilde{z}}_{2} \end{array}$$
where(12)$$\begin{array}{c} z_{1}-v_{0}=\lambda_{0}L^{\frac{1}{3}}{\left|z_{0}-x_{2}\right|}^{\frac{2}{3}}\mathrm{sign}\left(z_{0}-x_{2}\right)\\ =\lambda_{0}L^{\frac{1}{3}}{\left|{\tilde{z}}_{0}\right|}^{\frac{2}{3}}\mathrm{sign}\left({\tilde{z}}_{0}\right)\\ ={\tilde{z}}_{1}-{\dot{\tilde{z}}}_{0} \end{array}$$

Thus, (Equation 12) can be rewritten as ${\dot{\tilde{z}}}_{1}=-\lambda_{1}L^{\frac{1}{2}}{\left|{\tilde{z}}_{1}-{\dot{\tilde{z}}}_{0}\right|}^{\frac{1}{2}}\mathrm{sign}\left({\tilde{z}}_{1}-{\dot{\tilde{z}}}_{0}\right)+{\tilde{z}}_{2}$.

Furthermore, ${\dot{\tilde{z}}}_{2}={\dot{z}}_{2}-\ddot{f}=-\lambda_{2}L\mathrm{sign}\left(z_{2}-v_{1}\right)-\ddot{f}$, due to $z_{2}-v_{1}={\tilde{z}}_{2}-{\dot{\tilde{z}}}_{1}$, ${\dot{\tilde{z}}}_{2}=-\lambda_{2}L\mathrm{sign}\left({\tilde{z}}_{2}-{\dot{\tilde{z}}}_{1}\right)-\ddot{f}$.

Subsequently,(13)$$\begin{array}{c} {\dot{\tilde{z}}}_{0}=-\lambda_{0}L^{\frac{1}{3}}{\left|{\tilde{z}}_{0}\right|}^{\frac{2}{3}}\mathrm{sign}\left({\tilde{z}}_{0}\right)+{\dot{\tilde{z}}}_{1}\\ {\dot{\tilde{z}}}_{1}=-\lambda_{1}L^{\frac{1}{2}}{\left|{\tilde{z}}_{1}-{\dot{\tilde{z}}}_{0}\right|}^{\frac{1}{2}}\mathrm{sign}\left({\tilde{z}}_{1}-{\dot{\tilde{z}}}_{0}\right)+{\dot{\tilde{z}}}_{2}\\ {\dot{\tilde{z}}}_{2}=-\lambda_{2}L\mathrm{sign}\left({\tilde{z}}_{2}-{\dot{\tilde{z}}}_{1}\right)-\ddot{f} \end{array}$$

Considering Lemma 1, Equation (Equation 11) is finite-time stable, and the estimation errors ${\tilde{z}}_{0}$, ${\tilde{z}}_{1}$, and ${\tilde{z}}_{2}$ will converge to zero in finite time.

Subsequently, leveraging the designed FDO, an NFITSMC is developed. Firstly, the tracking error of the EHASS is defined as $e=x_{1}-x_{d}$. The derivative of *e* yields $\dot{e}=x_{2}-{\dot{x}}_{d}$.

To ensure position tracking accuracy, the nonsingular fast integral terminal sliding mode surface is designed as(14)$$s=\dot{e}+\int_{0}^{t}(\sigma_{2}{\left|\dot{e}\right|}^{\alpha_{2}}\mathrm{sign}(\dot{e})+\sigma_{1}{\left|e\right|}^{\alpha_{1}}\mathrm{sign}(e))d\tau$$
where $\sigma_{1}$ and $\sigma_{2}$ are positive constants, and $0<\alpha_{1}<1$, $\alpha_{2}=2\alpha_{1}/\left(\alpha_{1}+1\right)$. Then the derivative of *s* is given by(15)$$\begin{array}{c} \dot{s}=\ddot{e}+\sigma_{2}{\left|\dot{e}\right|}^{\alpha_{2}}\mathrm{sign}(\dot{e})+\lambda_{1}{\left|e\right|}^{\alpha_{1}}\mathrm{sign}(e)\\ =f+b_{0}u-{\ddot{x}}_{d}+\sigma_{2}{\left|\dot{e}\right|}^{\alpha_{2}}\mathrm{sign}(\dot{e})+\sigma_{1}{\left|e\right|}^{\alpha_{1}}\mathrm{sign}(e) \end{array}$$

The observation value of *f* is substituted into the sliding mode surface, and the exponential convergence law of sliding mode control is defined as $\dot{s}=-k_{1}s-k_{2}sgn\left(s\right)$; then the final control variable *u* can be derived as(16)$$u=\frac{1}{b_{0}}\left(-k_{1}s-k_{2}\mathrm{sign}\left(s\right)+{\ddot{x}}_{d}-z_{1}-\sigma_{2}{\left|\dot{e}\right|}^{\alpha_{2}}\mathrm{sign}(\dot{e})-\sigma_{1}{\left|e\right|}^{\alpha_{1}}\mathrm{sign}(e)\right)$$

In order to illustrate the stability of the control system, the candidate Lyapunov function is given by $V=\frac{1}{2}s^{2}$. Then the time derivative of *V* yields(17)$$\begin{array}{c} \dot{V}=s\dot{s}\\ =s(f+b_{0}u-{\ddot{x}}_{d}+\sigma_{2}{\left|\dot{e}\right|}^{\alpha_{2}}\mathrm{sign}(\dot{e})+\sigma_{1}{\left|e\right|}^{\alpha_{1}}\mathrm{sign}(e))\\ =s(-k_{1}s-k_{2}\mathrm{sign}\left(s\right)+f-z_{1}) \end{array}$$

From above, the estimation error ${\tilde{z}}_{1}=z_{1}-f$ converges to zero in finite time. Then, (Equation 18) can be rewritten as(18)$$\begin{array}{c} \dot{V}=s\left(-k_{1}s-k_{2}\mathrm{sign}\left(s\right)\right)\\ \leq -k_{1}s^{2}-k_{2}\left|s\right|\\ \leq -2k_{1}V-\sqrt{2}k_{2}V^{\frac{1}{2}} \end{array}$$

Now from (Equation 18) and Lemma 2, it can be concluded that the position tracking error will reach the sliding mode surface s=0 in finite time. When s=0, we have(19)$$\dot{e}=-\int_{0}^{t}(\sigma_{2}{\left|\dot{e}\right|}^{\alpha_{2}}\mathrm{sign}(\dot{e})+\sigma_{1}{\left|e\right|}^{\alpha_{1}}\mathrm{sign}(e))d\tau$$

Taking the derivative of (Equation 19), we can obtain(20)$$\ddot{e}=-\sigma_{2}{\left|\dot{e}\right|}^{\alpha_{2}}\mathrm{sign}(\dot{e})-\sigma_{1}{\left|e\right|}^{\alpha_{1}}\mathrm{sign}(e)$$

Considering Lemma 3, *e* and $\dot{e}$ will converge to zero in finite time.

## 4. Simulation

In this section, we conduct a numerical comparative simulation analysis of the three-axle vehicle suspension system to verify the feasibility of the proposed controller under the input of a bumpy road surface. To capture more realistic road scenarios and vehicle dynamics, we perform joint simulations using professional vehicle simulation software, Trucksim 2016, in conjunction with MATLAB 2018b. A three-axle heavy-duty vehicle model, incorporating real parameters, is simulated in Trucksim. The MEHASS model and the DCC are implemented in MATLAB, where the vertical tire force and control signals serve as transmission signals for real-time communication between the two software platforms.

The vehicle parameters and electro-hydraulic servo system parameters employed in the simulation are presented in Table 1. The stiffness coefficients and damping coefficients of the passive suspension are denoted by $k_{si}$ and $b_{si}$, respectively, for comparison purposes.

To simulate more realistic road conditions, this study integrates ISO-compliant random pavement with bumpy pavement to generate the pavement inputs illustrated in Figure 5. The vehicle’s driving speed is set to 10 km/h, with the height of the bumpy pavement inputs for the left and right wheels set at 0.08 m and 0.11 m, respectively. In the simulations, the DCC parameters comprise both generalized impedance parameters and position control parameters. Taking the active suspension subsystem for the left front wheel as an example, its generalized impedance parameters are defined as follows: $K_{d1}=3000$, $B_{d1}=80$, $K_{f1}=5$, $K_{I1}=1$. Position control parameters are assigned as $L=10,\;\lambda_{0}=4,\;\lambda_{1}=8,\;\lambda_{2}=12,\;\alpha_{1}=\frac{1}{2},\;\alpha_{2}=\frac{2}{3},\;\sigma_{1}=9,\;\sigma_{2}=6,\;k_{1}=2.5,\;k_{2}=1.6$. The parameters of the other suspension subsystems have been slightly adjusted compared to the above example, and the overall difference is not significant.

The simulation results are presented in Figure 6 and Figure 7. Figure 6 illustrates the vehicle motions and acceleration responses in the vertical, pitch, and roll directions, while Figure 7 depicts the variation in vertical forces for each wheel of the vehicle. Analyzing these simulation results reveals that under the input of a convex road surface, the proposed DCC method for the active suspension system demonstrates superior performance compared to the passive suspension system. It effectively reduces the fluctuation peaks in vertical displacement, pitch angle, roll angle, and their corresponding accelerations of the vehicle body. Additionally, it mitigates the dynamic load changes experienced by each wheel. These findings indicate that the proposed DCC method excels in balancing the vehicle’s load distribution, suppressing changes in vehicle body posture, and enhancing ride comfort when encountering uneven road surfaces.

## 5. Experiments

This section details the experimental vehicle prototype and suspension control system. Two case studies of road experiments are conducted to validate the effectiveness of the proposed DCC method.

### 5.1. Prototype Vehicle and Suspension Control System

The prototype vehicle and suspension control system are depicted in Figure 8 and Figure 9, respectively. The electro-hydraulic servo system of the experimental prototype vehicle was modified from the original HPS, which can achieve mutual switching between HPS and electro-hydraulic servo active suspension. Each wheel’s electro-hydraulic servo system comprises a servo actuator, an electro-hydraulic servo valve (SFD234, Aerospace JUNHE Technology Co., Ltd., Beijing, China), a servo control module, a displacement sensor (MHE0350MN14H3A01, XCMG construction machinery Limited by Share Ltd., Xuzhou, China), and two oil pressure sensors (PM95, TUUMU, Detroit, MI, USA). The servo control module for each wheel is designed based on stm32f373(STMicroelectronics, Plan-les-Ouates, Switzerland) and integrates a signal acquisition module and a CAN bus communication module. The servo control module collects signals from various sensors and generates current control signals to actuate the electro-hydraulic servo valve. Each servo control module is equipped with a built-in compliance control method, enabling decentralized control. The vehicle status acquisition module consists of an inertial measurement unit (MTi-300 AHRS, Xsens, Enschede, The Netherlands) installed near the vehicle centroid and an industrial personal computer (PC104, Shengbo Technology, Beijing, China). The industrial personal computer communicates with the inertial measurement unit via the serial port and with each servo control module through the CAN bus, facilitating monitoring of the status of each electro-hydraulic servo actuator system and the vehicle, as well as the collection of experimental data. The CAN bus communication operates at a baud rate of 500k, with a communication cycle of 4 ms, and the control cycle of each servo control module is also set to 4 ms.

### 5.2. Experimental Methods Related Parameters Setup

It is challenging to directly measure the real-time external contact force between each wheel and the uneven road surface. Instead, the output force of the actuator corresponding to each wheel can be calculated based on the real-time measured oil pressure values of the non-rod chamber and rod chamber. Moreover, since the actuator is rigidly connected to the wheel, its output force can to some extent reflect the variations in the contact force between the wheel and the road surface. Therefore, the output force of the actuator will be employed as the external contact force of the sprung mass for the practical implementation of the DCC during the experiment. The DCC is designed to maintain the static load force on the spring. Set $e_{fi}=P_{i1}A_{1}-P_{i2}A_{2}-m_{si}g$, where $m_{si}$ is the sprung mass carried by each actuator in the static equilibrium state of the vehicle, and $m_{si}g=55,000$ N, which represents the expected static output force of each. The DCC scheme based on the actuator output force is formulated as follows:(21)$$B_{di}\left({\dot{z}}_{di}-{\dot{z}}_{ri}\right)+K_{di}\left(z_{di}-z_{ri}\right)=-K_{fi}e_{fi}-K_{Ii}\int_{0}^{t}e_{fi}d_{t}$$

The parameters of the DCC are composed of generalized impedance parameters and position control parameters. Taking the active suspension subsystem corresponding to the left front wheel as an example, its generalized impedance parameters were $K_{d1}=4000$, $B_{d1}=100$, $K_{f1}=10$, $K_{I1}=0.5$ and its integral sliding mode parameters were $L=10,\;\lambda_{0}=4$, $\lambda_{1}=8,\;\lambda_{2}=12,\;\alpha_{1}=\frac{1}{2},\;\alpha_{2}=\frac{2}{3},\;\sigma_{1}=9,\;\sigma_{2}=6,\;k_{1}=1.5,\;k_{2}=1.2$. The parameters of the other suspension subsystems have been slightly adjusted compared to the above example, and the overall difference is not significant.

To verify the effectiveness of the proposed DCC method, comparative experiments were conducted using the HPS and the multi-objective PID (M-PID) method as benchmarks. The M-PID method, as referenced in [35], employs traditional centralized attitude control strategies to minimize the vehicle’s pitch and roll angles. In this study, the M-PID method is a comparative benchmark to evaluate the proposed DCC method against traditional centralized attitude control approaches. The parameters of the M-PID controller are set to $K_{p}^{\theta }=0.2,K_{I}^{\theta }=0.01;K_{p}^{\varphi }=0.15,K_{I}^{\varphi }=0.01$.

### 5.3. Comparative Experiments

*Case study on convex road*: Convex triangular obstacles, as depicted in Figure 10, were chosen for this experiment. The obstacle had a height of 0.1 m, a length of 1.0 m, and a width of 0.5 m. They were positioned adjacent to each other on a relatively level road surface, considering the distance between the wheels on both sides of the experimental vehicle. This arrangement ensures that the wheels on both sides of the vehicle can simultaneously traverse the obstacles. As the experiment involves the simultaneous crossing of obstacles using wheels on both sides of the vehicle, the resulting changes in pitch and vertical motion have a more pronounced impact on the vehicle’s attitude than the roll motion, which exhibits relatively smaller changes. Therefore, real-time data collection focused on the pitch angle and centroid vertical acceleration of the vehicle for comparative analysis. Furthermore, to assess the flexibility performance of the vehicle’s suspension systems under different control methods, information on the actuator output force and displacement for each wheel was compared and analyzed. For research purposes, the left and right front wheels of the vehicle’s front axle were selected as examples.

Figure 11 and Figure 12 present experimental data obtained at a vehicle speed of 7 km/h. Figure 11 illustrates the suspension actuators’ output force as the vehicle crosses a convex obstacle. It can be observed that the output force initially increases, then decreases, and eventually stabilizes. Comparatively, the DCC method exhibits smaller fluctuations in output force compared to the HPS and M-PID methods. As shown in Figure 12, the MPID method achieves a smaller peak pitch angle due to its direct control of the vehicle’s attitude. However, the DCC method demonstrates a smaller peak vertical acceleration value than the HPS and M-PID methods.

Figure 13 and Figure 14 illustrate the experimental data obtained at a vehicle speed of 10 km/h. It can be observed from Figure 13 that as the vehicle speed increases, the DCC exhibits smaller fluctuation peaks and amplitudes in the output force when compared to the HPS. Additionally, it shows a smaller deviation from the expected static output force. Figure 14 depicts the experimental comparison curves of the pitch angle and vertical acceleration of the vehicle body at a speed of 10 km/h. As observed in Figure 14, the DCC demonstrates superior performance over the HPS and M-PID, resulting in smaller fluctuation peaks and amplitudes in vertical acceleration of the vehicle body.

The peak-to-peak (PTP) and root mean square (RMS) values of the vehicle’s vertical acceleration were determined to statistically examine the performance improvement of the suggested DCC method on vehicle suspension control. Table 2 reveals that the DCC method shows smaller PTP values and RMS values of the vehicle’s vertical acceleration at both vehicle speeds. Although the M-PID method achieves smaller pitch angle PTP values, it exhibits larger actuator output force PTP values. This results in a higher peak vehicle vertical acceleration compared to the DCC method. The DCC method shows smaller actuator output force PTP values and peak vertical acceleration values at both vehicle speeds. This indicates that the DCC method is more effective in mitigating the transmission of road-roughness-induced forces to the chassis, thereby enhancing its ability to isolate impacts from uneven road surfaces.

*Case study on countryside unpaved road*: To further validate the effectiveness of the proposed method, road experiments were conducted on an unpaved countryside road, as illustrated in Figure 15, with a test speed of 30 km/h. Three control methods, HPS, M-PID, and the proposed DCC, were employed, and the experimental results are presented in Figure 16, Figure 17 and Figure 18. The figures demonstrate that both the DCC and M-PID methods exhibit superior attitude retention performance compared to HPS. However, the DCC method shows a smaller PTP actuator output force compared to both the M-PID and HPS methods. As shown in Figure 18 and Table 3, the DCC approach also results in reduced peak vertical acceleration and lower RMS values of body acceleration. These findings indicate that the DCC method significantly improves the ride comfort of the three-axle vehicle.

In comparative experiments conducted in two different road environments, the DCC method demonstrated a more stable actuator output force fluctuation amplitude and smaller PTP and RMS values of vehicle vertical acceleration compared to the HPS and M-PID methods. Therefore, it can be concluded that the DCC method effectively reduces the transmission of vehicle chassis vibrations caused by road surface disturbances, thereby improving the ride comfort of heavy-duty vehicles.

## 6. Conclusions

In this paper, a DCC method is proposed for the electro-hydraulic servo active suspension of a three-axle heavy vehicle. The DCC method involves two main components. Firstly, the generalized impedance rule is utilized to generate the vertical compliant trajectory of the electro-hydraulic actuators, which are rigidly connected to the wheel. Additionally, the NFITSMC is employed to track the calculated vertical compliant displacement of the actuators.

To validate the effectiveness of the proposed method, experimental research is conducted using an established experimental platform of a three-axle heavy vehicle equipped with electro-hydraulic servo active suspension. An engineering application experimental method of DCC is proposed and implemented. The experimental results are analyzed, leading to the following conclusions: The DCC method effectively reduces the transmission of vehicle chassis vibrations caused by road surface disturbances, thereby improving the ride comfort of heavy-duty vehicles. The proposed DCC method, with its decentralized structure, demonstrates greater suitability for three-axle and multi-axle heavy vehicles compared to traditional nonlinear control laws. This highlights its promising potential for widespread engineering applications. Future work will focus on the simultaneous enhancement of ride comfort and handling stability for multi-axle heavy-duty vehicles.

## Figures and Tables

**Figure 1 sensors-25-05456-f001:**
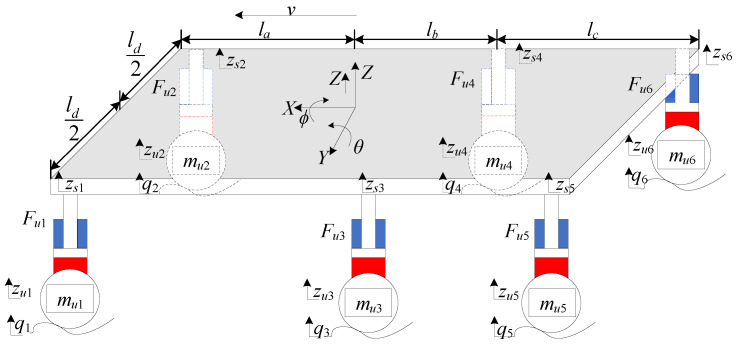
Three-axle heavy vehicle dynamic model.

**Figure 2 sensors-25-05456-f002:**
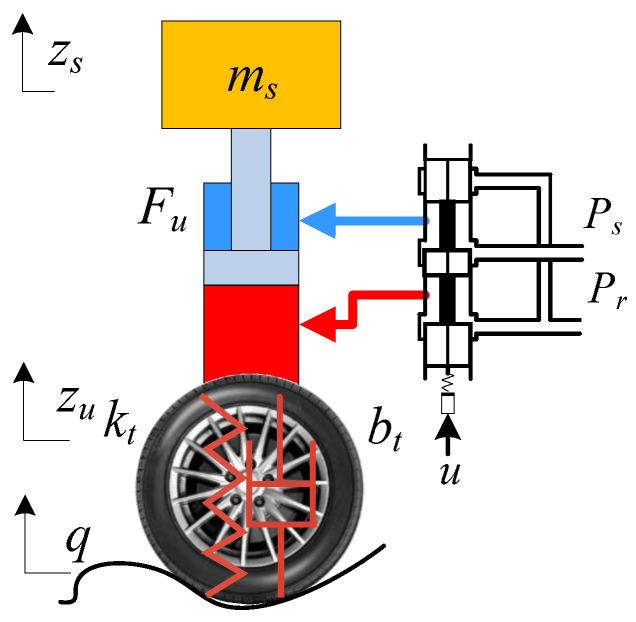
Schematic diagram of the EHASS.

**Figure 3 sensors-25-05456-f003:**
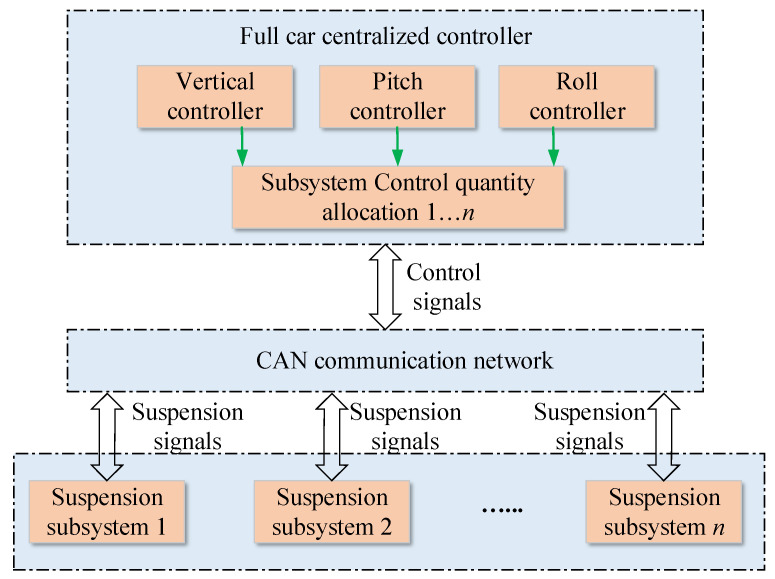
Schematic of a typical full car control scheme.

**Figure 4 sensors-25-05456-f004:**
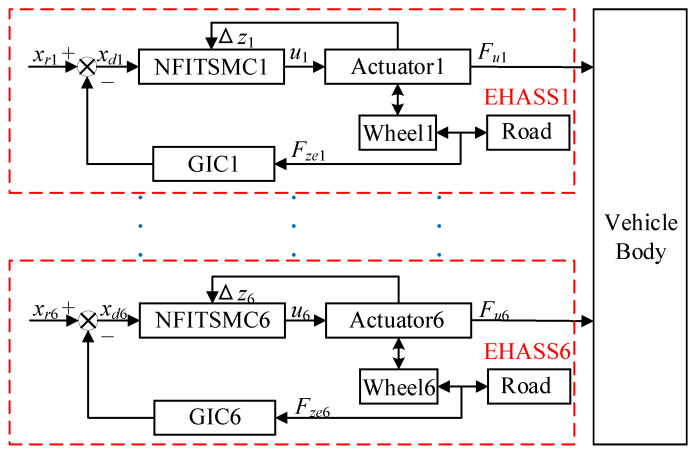
The DCC framework.

**Figure 5 sensors-25-05456-f005:**
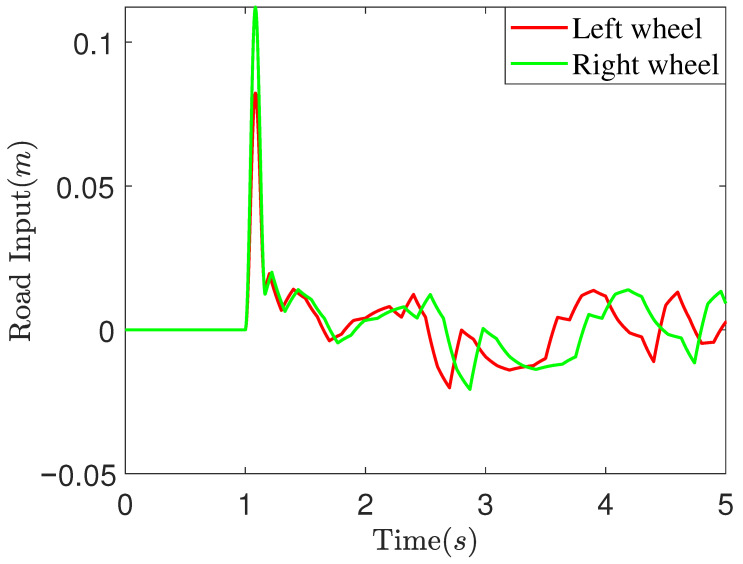
Bumpy road input.

**Figure 6 sensors-25-05456-f006:**
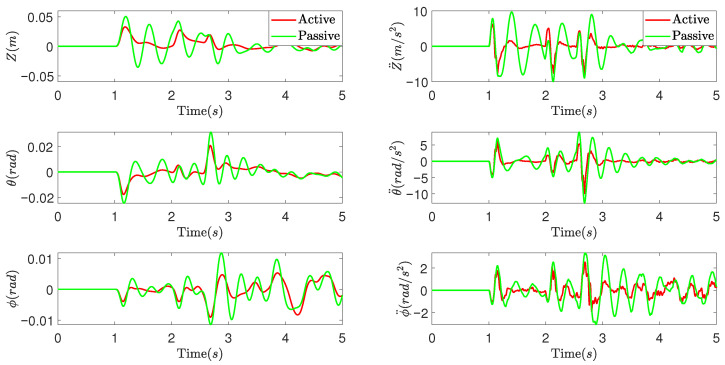
The vehicle motions and acceleration responses of the vehicle.

**Figure 7 sensors-25-05456-f007:**
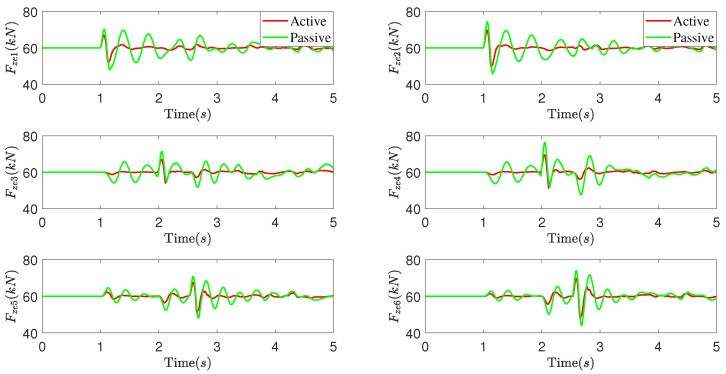
The variation in vertical forces for each wheel of the vehicle.

**Figure 8 sensors-25-05456-f008:**
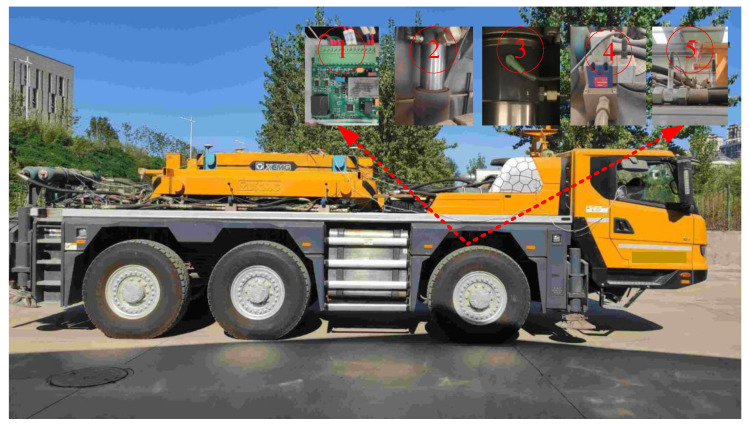
The three-axle heavy vehicle prototype: The electro-hydraulic servo system corresponding to the right front wheel: (1) shows the servo controller, (2) shows the hydraulic actuator, (3) illustrates the displacement sensor, (4) shows the servo valve, and (5) shows the oil pressure sensors.

**Figure 9 sensors-25-05456-f009:**
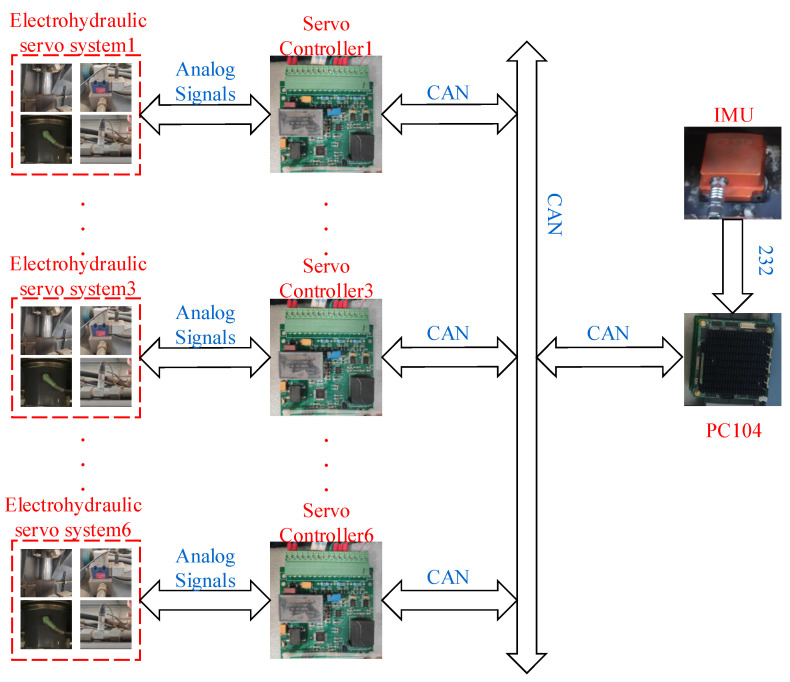
The suspension control system of the prototype vehicle.

**Figure 10 sensors-25-05456-f010:**
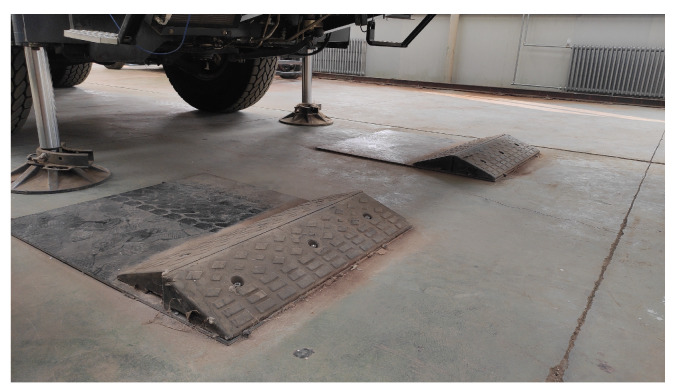
On-road driving experiment with the convex triangular obstacles.

**Figure 11 sensors-25-05456-f011:**
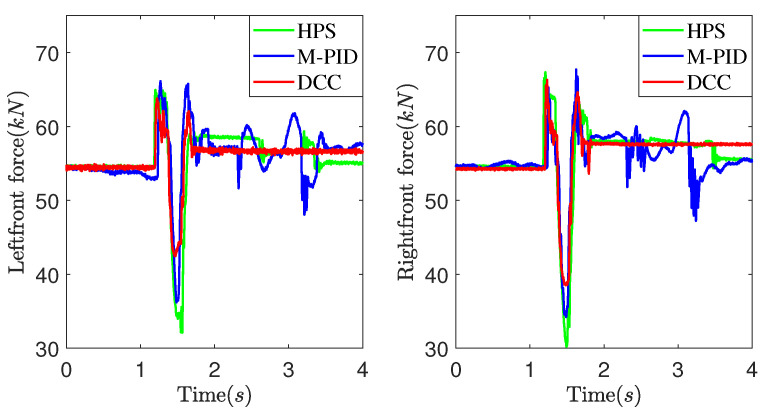
Suspension actuators output force at 7 km/h.

**Figure 12 sensors-25-05456-f012:**
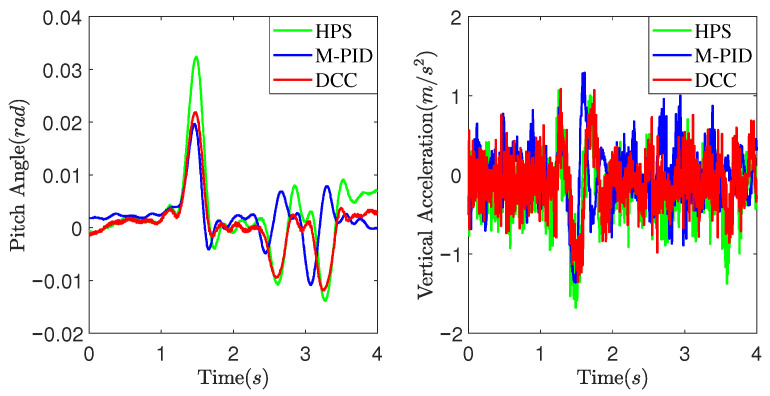
The pitch angle and vertical acceleration of the vehicle body at 7 km/h.

**Figure 13 sensors-25-05456-f013:**
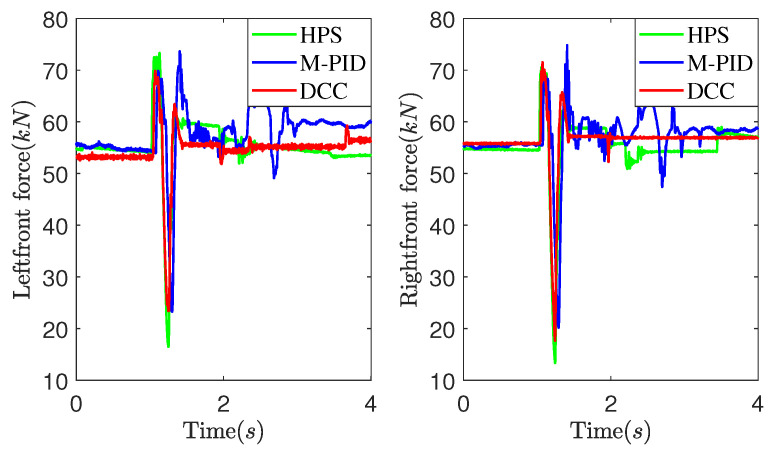
Suspension actuators output force at 10 km/h.

**Figure 14 sensors-25-05456-f014:**
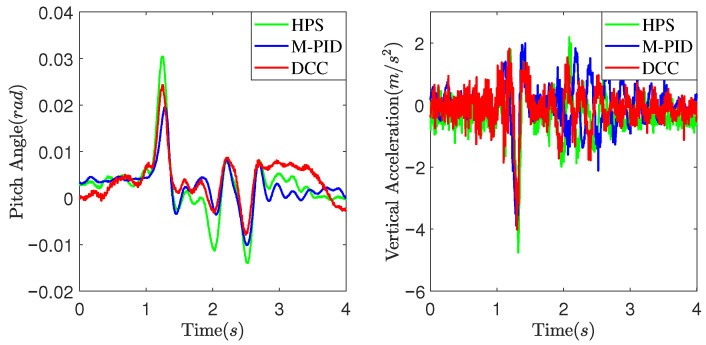
The pitch angle and vertical acceleration of the vehicle body at 10 km/h.

**Figure 15 sensors-25-05456-f015:**
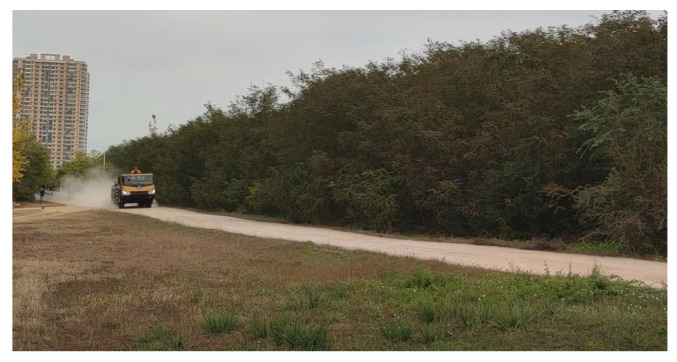
On-road driving experiment with the countryside unpaved road.

**Figure 16 sensors-25-05456-f016:**
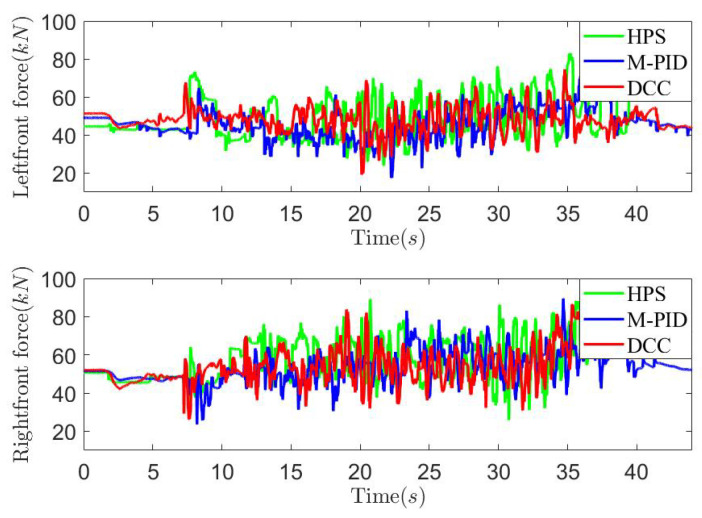
Left front and right front actuator output force curves on the countryside unpaved road.

**Figure 17 sensors-25-05456-f017:**
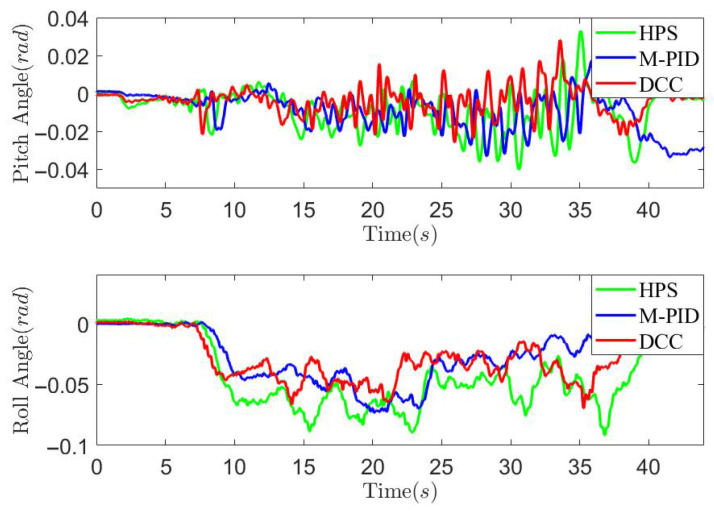
Vehicle pitch angle and roll angle curves on the countryside unpaved road.

**Figure 18 sensors-25-05456-f018:**
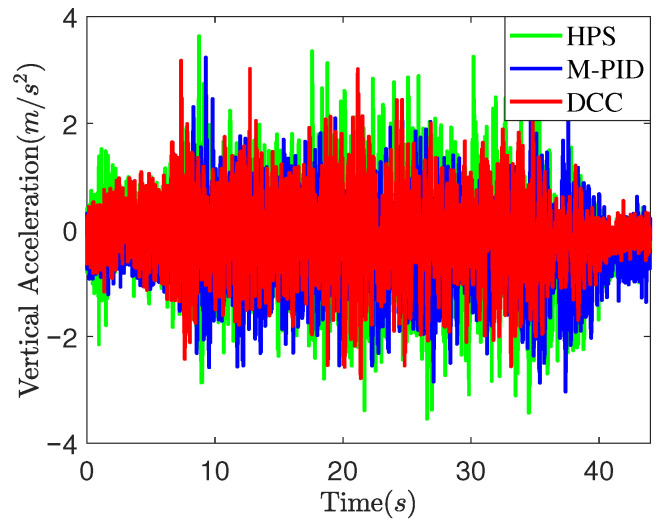
Vehicle vertical acceleration curves on the countryside unpaved road.

**Table 1 sensors-25-05456-t001:** Parameters of the vehicle and the MEHASS.

Parameters	Values	Parameters	Values
*M*	33,000 kg	$k_{si}$	150,000 N/m
$m_{si}$	5500 kg	$b_{si}$	100,000 N/(m/s)
$m_{ui}$	500 kg	$A_{1}$	0.00567 $m^{2}$
$l_{a}$	1.8875 m	$A_{2}$	0.00125 $m^{2}$
$l_{b}$	1.0625 m	$P_{s}$	$2.5\times 10^{7}$ Pa
$l_{c}$	1.65 m	$V_{01}$	$1.1391\times 10^{-3}$ $m^{3}$
$l_{d}$	2.125 m	$V_{02}$	$0.1239\times 10^{-3}$ $m^{3}$
$I_{x}$	26,506 kg $m^{2}$	$\beta_{e}$	$2\times 10^{8}$ N $m^{2}$
$I_{y}$	78,593 kg $m^{2}$	$C_{t}$	$4.5\times 10^{-15}$
$k_{ti}$	1,900,000 N/m	$k_{q}$	$1.019\times 10^{-7}$
$b_{ti}$	15,000 N/(m/s)		

**Table 2 sensors-25-05456-t002:** The comparison data for vertical acceleration of the three methods under different vehicle speeds.

Acceleration	HPS	M-PID	DCC
PTP value (7 km/h)	2.747	2.649 (↓3.6%)	2.435 (↓11.3%)
RMS value (7 km/h)	0.452	0.396 (↓12.4%)	0.387 (↓14.4%)
PTP value (10 km/h)	6.94	6.04 (↓12.9%)	5.85 (↓15.8%)
RMS value (10 km/h)	0.81	0.77 (↓4.9%)	0.64 (↓21.0%)

Note: The % variations are with respect to the HPS.

**Table 3 sensors-25-05456-t003:** The comparison data for vertical acceleration of the three methods on the countryside unpaved road.

Acceleration	HPS	M-PID	DCC
PTP value	7.17	6.26 (↓12.7%)	5.95 (↓17.0%)
RMS value	0.897	0.816 (↓9.1%)	0.739 (↓17.6%)

Note: The % variations are with respect to the HPS.

## Data Availability

The original contributions presented in this study are included in the article. Further inquiries can be directed to the corresponding author.

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
