# Peer review of "Decentralized Compliance Control for Multi-Axle Heavy Vehicles Equipped with Electro-Hydraulic Actuator Suspension Systems"

_sensors, 2025, doi:10.3390/s25175456_

Round 1

Reviewer 1 Report

Comments and Suggestions for Authors

This research paper designs a controller for active suspension
systems for multi-axle heavy vehicles, the decentralized scheme proposed in this paper decomposes the overall vehicle control problem into decentralized compliance control tasks for multiple electro-hydraulic actuator suspension subsystems. Simulation and experimental test are conducted. Some comments should be checked and considered.
1. The description sentences said the right bump pavement input is 0.1m,however, in Fig. 5, the input of right wheel is larger than 0.1m,please explain.
2. It is recommended to use different linestyles, e.g., Figs. 16, 17, 18.
3. Some newest technologies about suspension and control strategy should be discussed, e.g., 10.1007/s10409-023-23367-x.
4. Most of the references in this research paper are old, it is recommended to add some new researh about vehicle suspension in the last three years.
5. Are there references of the vehicle parameters? And regarding the MEHASS, are the parameters optimized?
6. Does the model in this research consider the tire damping? In Figure 2 and Eq. 1, it has, while in Table 1, this parameter is not given.

Author Response

Comments 1: The description sentences said the right bump pavement input is 0.1m,however, in Fig. 5, the input of right wheel is larger than 0.1m,please explain.

Response 1: Thanks for your reminder, which has contributed to enhancing the quality of our manuscript. The road input selected in the simulation environment of the manuscript is a combination of a bump road surface and a random road surface. Consequently, the road input height in the figure exceeds that of the bump-only road input (0.11 m).

Comments 2: It is recommended to use different linestyles, e.g., Figs. 16, 17, 18.

Response 2: Thanks for your suggestions. In the manuscript, all experimental curves maintain consistent line-type coding: green corresponds to HPS, blue to M-PID, and red to the proposed DCC method. Despite being derived under varying road excitations, this uniform representation significantly enhances reader comprehension.

Comments 3: Some newest technologies about suspension and control strategy should be discussed, e.g., 10.1007/s10409-023-23367-x.

Response 3: Thanks for your reminder, which has contributed to enhancing the quality of our manuscript. Relevant discussions on new literature and their control methods, along with corresponding citations, have been incorporated into the Introduction section. Specific modifications are detailed below:

“[In recent years, significant progress has been made in the field of advanced nonlinear control for electro-hydraulic actuator active suspension systems. Various control approaches, including multi-objective control[13], adaptive control[14], and prescribed performance control[15], have been applied to active suspension, addressing challenges such as parameter uncertainty, nonlinear damping, and nonlinear actuator behavior. Zheng proposed a backstepping approach for controlling the active suspension of a seven degrees-of-freedom(DOF) nonlinear full-vehicle model[16]. Na introduced a prescribed performance control method that effectively handles uncertainties and nonlinearities in the active suspension system without relying on function approximators[17]. Dahunsi adopted multi-loop PID controllers for full-car nonlinear electrohydraulic active suspension system to resolve the conflicting performance criteria[18].]”

Comments 4: Most of the references in this research paper are old, it is recommended to add some new researh about vehicle suspension in the last three years.

Response 4: Thanks for your suggestions. Relevant discussions on new literature and their control methods, along with corresponding citations, have been incorporated into the Introduction section.

Comments 5: Are there references of the vehicle parameters? And regarding the MEHASS, are the parameters optimized?

Response 5: Thanks for your suggestions. The vehicle parameters used in the manuscript’s simulations were sourced from the experimental prototype’s real-world data, including manufacturer-specified values and measured values. Since the proposed control method is robust to vehicle parameters, no optimization algorithms were applied to fine-tune them.

Comments 6: Does the model in this research consider the tire damping? In Figure 2 and Eq. 1, it has, while in Table 1, this parameter is not given.

Response 6: Thanks for your reminder, which has contributed to enhancing the quality

of our manuscript. Tire damping parameters involved in the simulations have now been incorporated into the manuscript.

Reviewer 2 Report

Comments and Suggestions for Authors
  1. This article proposes a decentralized compliance control (DCC) method for the electro-hydraulic actuator suspension system of a three-axle heavy-duty vehicle. This method is innovative and has potential for engineering application.
  2. Figures 3 and 4 show differences of the proposed method that does not require the transmission of each wheel's status to the same CAN for integration. Does this method enhance the immediate response of each axle? If one wheel lacks immediate response, does it affect the stability of the entire vehicle?
  3. This method appears to help maintain vehicle stability/comfort when the vehicle is traveling straight, but it appears to have a negative impact on vehicle cornering requirements. Should this be explained?

Author Response

Comments 1: This article proposes a decentralized compliance control (DCC) method for the electro-hydraulic actuator suspension system of a three-axle heavy-duty vehicle. This method is innovative and has potential for engineering application.

Response 1: We sincerely appreciate the reviewers' recognition of the manuscript's innovativeness, which is of great significance to our research.

Comments 2: Figures 3 and 4 show differences of the proposed method that does not require the transmission of each wheel's status to the same CAN for integration. Does this method enhance the immediate response of each axle? If one wheel lacks immediate response, does it affect the stability of the entire vehicle?

Response 2: We appreciate the reviewers' patient evaluation and constructive comments. The decentralized architecture proposed in the manuscript indeed offers advantages over the integrated architecture in terms of communication structure and speed. If a wheel loses response, it adversely affects overall vehicle stability—a limitation shared by both architectures. Incorporating fault-tolerant design can mitigate this issue.

Comments 3: This method appears to help maintain vehicle stability/comfort when the vehicle is traveling straight, but it appears to have a negative impact on vehicle cornering requirements. Should this be explained?

Response 3: We appreciate the reviewer's suggestion, which is crucial for improving the manuscript. The proposed method primarily targets ride comfort enhancement, while its focus on handling stability—particularly regarding cornering and braking performance—remains relatively underdeveloped. This limitation, acknowledged in the Conclusions section, will be addressed in our subsequent research.

Reviewer 3 Report

Comments and Suggestions for Authors

In this manuscript, a novel decentralized compliance control technique designed to manage the behavior of multi-axle heavy vehicles equipped with electro-hydraulic actuator suspension systems on uneven terrains is proposed. The reviewer has a few concerns that the authors may consider.

  1. Abbreviations are suggested before Section 1.
  2. In Eq. (6), how to determine the value of the integration terms? Please provide some discussion on this.
  3. The research background needs to be further strengthened. Advanced control strategies, such as those presented in ‘10.1016/j.ecmx.2025.101129’ and ‘10.1109/ACCESS.2021.3101038’, should be discussed to better highlight the innovation of this manuscript.
  4. In the simulation, it is better to give the system and control gain paramters? I would suggest the list of scenarios should be presented, what is the difference between these scenarios.
  5. I kindly request that author review its experimental section. A more comprehensive description of the laboratory setup is necessary, including details about the types of equipment used and the precision of the sensors (for replicability reasons). Additionally, please provide more detailed information on how the system was hardware-implemented. It is essential to explain the procedure of data acquisition and transfer of the plotted results to MATLAB. Including oscillograms captured during the experiments would enhance the credibility of your experimental work. Furthermore, we recommend a significant improvement in the quality of the provided plots.

Comments on the Quality of English Language

The English could be improved to more clearly express the research.

Author Response

Response 1: Thanks for your suggestions. The manuscript now includes a dedicated Abbreviations section, as detailed below.

DCC

decentralized compliance control

MEHASS

multiple electro-hydraulic actuators suspension subsystems

NFITSMC

nonsingular fast integral terminal sliding mode controller

GIC

generalized impedance controller

HPS

hydro-pneumatic suspension

EHASS

electro-hydraulic actuator suspension subsystem

DOF

degrees-of-freedom

PTP

peak-to-peak

RMS

root-mean-square

M-PID

multi-objective PID

Comments 2: In Eq. (6), how to determine the value of the integration terms? Please provide some discussion on this.

Response 2: We appreciate the reviewers' patient evaluation and constructive comments. In Equation 6, f denotes the lumped disturbance term, which incorporates nonlinear components of control gains, modeling uncertainties, and external disturbances. A disturbance observer is utilized to estimate this lumped term, and a disturbance rejection component is integrated into the controller design.

Comments 3: The research background needs to be further strengthened. Advanced control strategies, such as those presented in ‘10.1016/j.ecmx.2025.101129’ and ‘10.1109/ACCESS.2021.3101038’, should be discussed to better highlight the innovation of this manuscript.

Response 3: Thanks for your reminder, which has contributed to enhancing the quality

of our manuscript. Relevant discussions on new literature and their control methods, along with corresponding citations, have been incorporated into the Introduction section. Specific modifications are detailed below:

“[In recent years, significant progress has been made in the field of advanced nonlinear control for electro-hydraulic actuator active suspension systems. Various control approaches, including multi-objective control[13], adaptive control[14], and prescribed performance control[15], have been applied to active suspension, addressing challenges such as parameter uncertainty, nonlinear damping, and nonlinear actuator behavior. Zheng proposed a backstepping approach for controlling the active suspension of a seven degrees-of-freedom(DOF) nonlinear full-vehicle model[16]. Na introduced a prescribed performance control method that effectively handles uncertainties and nonlinearities in the active suspension system without relying on function approximators[17]. Dahunsi adopted multi-loop PID controllers for full-car nonlinear electrohydraulic active suspension system to resolve the conflicting performance criteria[18].]”

Comments 4: In the simulation, it is better to give the system and control gain paramters? I would suggest the list of scenarios should be presented, what is the difference between these scenarios.

Response 4: Thanks for your reminder, which has contributed to enhancing the quality

of our manuscript. System and control gain parameters have been incorporated into the simulation description.

Comments 5: I kindly request that author review its experimental section. A more comprehensive description of the laboratory setup is necessary, including details about the types of equipment used and the precision of the sensors (for replicability reasons). Additionally, please provide more detailed information on how the system was hardware-implemented. It is essential to explain the procedure of data acquisition and transfer of the plotted results to MATLAB. Including oscillograms captured during the experiments would enhance the credibility of your experimental work. Furthermore, we recommend a significant improvement in the quality of the provided plots.

Response 5: We appreciate the reviewer's suggestions, which significantly contribute to improving the manuscript quality. The displacement sensor employed in experiments is a magneto strictive sensor with a range of 0-220 mm and accuracy of 0.1 mm. The hydraulic pressure sensor utilizes a sputtered thin-film type, featuring a 0-40 MPa range and 0.5 MPa accuracy. As the hardware implementation of the servo controller falls beyond this study's scope, more detailed specifications cannot be provided. Data acquisition was performed via a PC104 industrial computer monitoring vehicle states through the CAN bus, with stored data subsequently exported to MATLAB for curve plotting. Non-standard data curves in the manuscript have been redrawn to comply with publication standards.

4. Response to Comments on the Quality of English Language

Response 1: Thanks for your suggestions. The syntax errors have been checked and corrected. The modifications are shown below:

Revise the seventh line of the sixth paragraph of section 1 " consists a generalized impedance " to" consists of a generalized impedance ".

Revise the eleventh line of the sixth paragraph of section 1 " A NFITSMC " to" The NFITSMC ".

Revise the second line of the first paragraph of section 3 " The force applied on the tire " to" The force applied to the tire".

Revise the seven line of the second paragraph of section 3 " A NFITSMC " to" An NFITSMC ".

Revise the third line of the second paragraph of section 3.1 " the vertical force applied on each wheel " to" the vertical force applied to each wheel ".

Revise the third line of the first paragraph of section 3.2 " for position control " to" for the position control ".

Reviewer 4 Report

Comments and Suggestions for Authors

In the paper the authors proposed a novel  mathematical model of electro-hydraulic servo active suspension of a three-axle heavy vehicle. Proposed approach effectively reduces the transmission of vehicle chassis vibrations caused by road surface disturbances, thereby improving the ride comfort of heavy-duty vehicles.

The results of the study are scientifically based, well presented and will be of interest to readers.

The paper can be accepted in present form.

Author Response

Overall Comment: In the paper the authors proposed a novel  mathematical model of electro-hydraulic servo active suspension of a three-axle heavy vehicle. Proposed approach effectively reduces the transmission of vehicle chassis vibrations caused by road surface disturbances, thereby improving the ride comfort of heavy-duty vehicles.

The results of the study are scientifically based, well presented and will be of interest to readers.

The paper can be accepted in present form.

Response to Overall Comment: We sincerely appreciate the reviewers' recognition of the manuscript's innovativeness, which is of great significance to our research.

Round 2

Reviewer 1 Report

Comments and Suggestions for Authors

Thanks for the authors' effort to reply to the queries or comments. But there still exist concerns. Some newest technologies about suspension and control strategy should be discussed, e.g., 10.1007/s10409-023-23367-x.  It is recommended to add some new researh about vehicle suspension in the last three years. 

Regarding the parameters of MEHASS, are the performance will be further enhanced if the parameters are optimized?

Author Response

Comments 1: Thanks for the authors' effort to reply to the queries or comments. But there still exist concerns. Some newest technologies about suspension and control strategy should be discussed, e.g., 10.1007/s10409-023-23367-x.  It is recommended to add some new researh about vehicle suspension in the last three years. 

Response 1: Thanks for your reminder, which has contributed to enhancing the quality of our manuscript. Relevant discussions on new literature and their control methods, along with corresponding citations, have been incorporated into the Introduction section. Specific modifications are detailed below:

“Yang proposed the concept of mechanical impedance to optimize suspension performance through frequency-dependent equivalent impedance methods; Zhang applied impedance control to full-vehicle active suspension systems, significantly enhancing ride comfort.”

Comments 2: Regarding the parameters of MEHASS, are the performance will be further enhanced if the parameters are optimized?

Response 2: The decentralized compliance control strategy adopted in the manuscript is robust to precise MEHASS parameters, eliminating the need for vehicle parameter optimization. However, control performance can be enhanced through tuning compliance control parameters, which constitutes a key focus of future work.

Reviewer 3 Report

Comments and Suggestions for Authors

I have no further comment

Comments on the Quality of English Language

no

Author Response

We sincerely appreciate the reviewers' recognition of the manuscript's innovativeness, which is of great significance to our research.